# Dietary Isorhamnetin Intake Is Associated with Lower Blood Pressure in Coronary Artery Disease Patients

**DOI:** 10.3390/nu14214586

**Published:** 2022-11-01

**Authors:** Joanna Popiolek-Kalisz, Piotr Blaszczak, Emilia Fornal

**Affiliations:** 1Clinical Dietetics Unit, Department of Bioanalytics, Medical University of Lublin, ul. Chodzki 7, 20-093 Lublin, Poland; 2Department of Cardiology, Cardinal Wyszynski Hospital in Lublin, al. Krasnicka 100, 20-718 Lublin, Poland; 3Department of Bioanalytics, Medical University of Lublin, ul. Jaczewskiego 8b, 20-090 Lublin, Poland

**Keywords:** flavonols, quercetin, hypertension, blood pressure, isorhamnetin

## Abstract

Background: Recent studies suggest the positive role of flavonols on blood pressure (BP) values, although there are not many conducted on humans. The aim of this study was to examine the relationship between flavonol intake and their main sources of consumption, and systolic (SBP) and diastolic (DBP) BP values in coronary artery disease (CAD) patients. Methods and results: forty CAD patients completed a food-frequency questionnaire dedicated to flavonol-intake assessment. The analysis revealed significant correlation between isorhamnetin intake and SBP values—absolute (R: −0.36; 95% CI: −0.602 to −0.052; *p* = 0.02), and related to body mass (R: −0.38; 95% CI: −0.617 to −0.076; *p* = 0.02. This effect was observed in male participants (R: −0.65; 95% CI: −0.844 to −0.302; *p* = 0.001 and R: −0.63; 95% CI: −0.837 to −0.280; *p* = 0.002 respectively), but not in female patients. The main contributors were onions, tomatoes, blueberries, apples, tea, coffee and wine. White onion (R: −0.39; 95% CI: −0.624 to −0.088; *p* = 0.01) consumption was inversely correlated with SBP, and tomato consumption (R: −0.33; 95% CI: −0.581 to −0.020; *p* = 0.04) with DBP. The comparison between patients with BP < 140 mmHg and ≥140 mmHg revealed significant differences in white onion (*p* = 0.01) and blueberry (*p* = 0.04) intake. Conclusions: This study revealed the relationship between long-term dietary isorhamnetin intake and SBP values. The analysis of specific food intake showed that onion, tomato and blueberry consumption could impact BP values. This may suggest that a dietary approach which includes a higher intake of isorhamnetin-rich products could possibly result in BP lowering in CAD patients.

## 1. Introduction

Elevated blood pressure (BP) is the leading cardiovascular disease (CVD) in Poland and it is present in 9.94 million adults, which is about 26% of total Polish population [1]. The overall prevalence of hypertension in adults is around 30–45% [2]. It is also a recognized cardiovascular risk factor, which is why finding a pattern leading to this condition is the main target of primary and secondary prevention. Apart from pharmacological treatment, which is introduced after a hypertension diagnosis, lifestyle changes including dietary approach are essential to prevent this condition, and as the first-line treatment [3]. The European Society of Cardiology recommendations for hypertension management include a diet rich in vegetables and fruit, although they are not very precise [3]. Vegetables and fruits are the sources of flavonoids which are investigated in varying contexts of human health, due to their antioxidative properties. Flavonols are the group of flavonoids distinguished by their chemical structure including a 3-hydroxyflavone backbone. They differ in the presence and position of hydroxyl and methyl groups. The main flavonols are quercetin, kaempferol, isorhamnetin and myricetin, although there is a large group of flavonols which are less abundant in the everyday diet, e.g., morin, galangin, fisetin, kaempferide, azaleatin, natsudaidain, pachypodol and rhamnazin [4,5,6]. The bioactivity of each compound depends on the number and type of functional groups. Quercetin, kaempferol and myricetin differ in the number of hydroxyl groups, while isorhamnetin is O-methylated in the R3 position, compared to quercetin.

The products particularly rich in flavonols are onions, tea, and apples, although kale, lettuce, tomatoes, broccoli, grapes, berries and red wine are also known to be flavonol-rich [6,7,8]. The main dietary contributors for quercetin intake are tomatoes, kale, apples and tea; for an intake of kaempferol, kale, beans, tea, spinach, and broccoli; for isorhamnetin, pears, olive oil, wine, and tomato sauce; for myricetin intake, tea, wine, kale, oranges, and tomatoes [9]. 

The most investigated flavonol is quercetin. The interventional studies in humans suggest the impact of its supplementation in BP regulation [10,11,12,13]. The other flavonols have not been the subject of interventional studies in humans yet, although the studies on animal models also suggest a positive role for isorhamnetin supplementation in hypertension management [14]. What is more, its potential role as a cardioprotective, neuroprotective, anti-tumor and anti-obesity agent was also suggested in in vitro and animal model studies [15,16,17,18]. Nonetheless, the results from the only observational human study which investigated the relationship between dietary-antioxidant habitual intake and hypertension are not consistent with this, as there was no observed correlation between flavonol (quercetin, kaempferol, isorhamnetin and myricetin) intake and hypertension incidence [7]. Nonetheless, it is worth noting that general hypertension diagnosis is based on crossing the limit of 140 mmHg for systolic BP (SBP) and/or 90 mmHg for diastolic BP (DBP) [3]. There has not yet been any study investigating the linear relationship between flavonol intake and BP values in coronary artery disease (CAD). 

The consumption of apples, which are a main source of flavonols, is generally advised in terms of health benefits (“an apple a day keeps a doctor away”); however, there have not been any studies which have analyzed the impact of apple consumption on BP values [19]. On the other hand, patients are often discouraged from drinking coffee, which is also a good source of flavonols, due to its potential negative impact on BP values, even though the recent studies do not confirm this [20].

The aim of this study was to analyze the impact of long-term dietary intake of the selected flavonols (quercetin, kaempferol, isorhamnetin and myricetin) and their main dietary sources, on levels of SBP and DBP among patients with CAD. Additionally, the impact of long-term consumption of apples and coffee, which are sources of flavonols, on BP values, was also investigated. 

## 2. Materials and Methods

Forty adult patients hospitalized between March and July 2022 due to CAD were enrolled in this study. Inclusion criteria were: (1) CAD diagnosis (2) age ≥ 18 years (3) written consent (4) mental condition that enabled a one-year retrospective dietary interview. The food-frequency questionnaire dedicated to specific flavonol one-year-intake assessment was administered to the patients [21]. On the basis of this, the mean daily intake for quercetin, kaempferol, isorhamnetin, myricetin and total flavonols was calculated for each patient. The information about mean daily intake of flavonol sources was also derived from the questionnaire. 

Patient weight was measured to 0.05 kg accuracy using the WTL-150A scale (Lubelskie Fabryki Wag), by a trained professional. The patient was allowed to wear only underwear for this measurement. BP was measured by a trained professional to 1 mmHg accuracy with the Omron M3 monitor (Omron Healthcare). The measurement was performed according to the European Society of Cardiology recommendations [22]. 

The study was approved by the local Bioethics Committee of the Medical University of Lublin (consent no. KE-0254/9/01/2022). The study was conducted in line with the directives of the Declaration of Helsinki on Ethical Principles for Medical Research. All participants signed a written consent agreement. 

Statistical analyses were performed with the RStudio software v. 4.2.0. The normality of the distribution of each parameter was checked by the Shapiro–Wilk test. The variables were presented as means (SD). Pearson correlation was used to analyze the association between selected flavonol mean daily-intake and SBP or DBP, and between selected products mean daily-intake and SBP or DBP. The cut-off points used for correlation coefficient were as follows: <0.20 as low, 0.20–0.49 as moderate and ≥0.50 as high correlation. A *p* value below 0.05 was considered significant. 

The patients were also divided into two groups due to SBP value—below 140 mmHg and 140 mmHg or higher. The differences in selected flavonol mean daily-intake and selected mean daily products between the groups were investigated by the Mann–Whitney test. A *p* value below 0.05 was considered significant.

## 3. Results

A total of 40 patients (21 men and 19 women) were enrolled in the study. The mean age was 68 (±9) years and mean weight was 80.40 (±10.30) kg. The mean daily total flavonol intake was 62.64 (±33.98) mg/day, and for specific flavonols: 29.77 (±22.18) mg/day for quercetin, 14.86 (±8.56) mg/day for kaempferol, 2.46 (±2.02) mg/day for isorhamnetin and 5.55 (±4.16) mg/day for myricetin. When the values were referred to body mass, the mean daily intake was 0.80 (±0.45) mg/kg for total flavonols, 0.51 (±0.29) mg/kg for quercetin, 0.19 (±0.11) mg/kg for kaempferol, 0.03 (±0.02) mg/kg for isorhamnetin and 0.07 (±0.05) mg/kg for myricetin. The mean measured SBP was 134.23 (±25.49) mmHg and DBP was 73.50 (±11.29) mmHg. The main contributors to flavonol intake were onions (white and red), tomatoes, blueberries, apples, tea, coffee and wine. 

The results revealed a significant moderate correlation between daily isorhamnetin intake and SBP values. The relation was present in both absolute daily intake (R: −0.36; 95% CI: −0.602 to −0.052; *p =* 0.02) and daily intake related to body mass (R: −0.38; 95% CI −0.617 to −0.076; *p =* 0.02). The detailed results for all analyzed flavonols are presented in Table 1.

The analysis of the male and female subgroup revealed that this effect was observed only in male participants (R: −0.65; 95% CI: −0.844 to −0.302; *p* = 0.001 for absolute isorhamnetin intake and R: −0.63; 95% CI: −0.837 to −0.280; *p* = 0.002 for related-to-body-mass isorhamnetin intake). The detailed results are presented in Table 2 and Table 3. 

The analysis of the main flavonol sources mean daily intake showed that onion (R: −0.38; 95% CI −0.616 to −0.074; *p* = 0.02) and white onion (R: −0.39; 95% CI: −0.624 to −0.088; *p =* 0.01) intake is correlated with SBP values, while tomato intake (R: −0.33; 95% CI: −0.581 to −0.020; *p =* 0.04) is correlated with DBP values. The detailed results for all analyzed products are presented in the Table 4.

The subgroup analysis revealed significant differences in isorhamnetin intake related to body mass (*p* = 0.048), white onions (*p =* 0.01) and blueberries (*p =* 0.04) among the patients with normal BP (<140 mmHg) and elevated BP (≥140 mmHg). The detailed results for all compounds and food are presented in the Table 5 and Figure 1. The additional analysis between the patients consuming less than 1 apple a day and ≥1 apple daily did not show any significant differences in terms of SBP (*p =* 0.55) or DBP (*p =* 0.95). A similar observation was made for coffee consumption (*p =* 0.64 for SBP and *p =* 0.43 for DBP).

## 4. Discussion

Flavonols are the subgroup of flavonoids which share a 3-hydroxyflavone backbone. Individual flavonols differ in their chemical structure (i.e., presence and position of hydroxyl and methyl groups) which impacts their bioactivity [23,24]. The most investigated flavonol is quercetin, and the studies in humans suggest the positive role of its supplementation in BP regulation [10,11,12,13]. The other flavonols are not as widely analyzed in this context. Most of the interventional studies about isorhamnetin impact on BP come from animal models, where isorhamnetin restored vasodilatation in hypertensive rats [14]. The mechanisms of this effect are still under investigation. However, the role of modification of protein kinases (C and Rho) activity, cytosolic H_2_O_2_ production or angiotensin-converting enzyme inhibition potential are suggested by other authors [14,25,26,27]. On the other hand, in the only observational human study which analyzed the impact of dietary antioxidant intake on hypertension, there was no correlation between general habitual dietary flavonoid intake and a reduction in incidents of hypertension [7]. A similar observation was also made in this study for specific flavonols (quercetin, kaempferol, isorhamnetin and myricetin) [7]. It is worth noting that the study was conducted on a very large population (a total of 156 957 participants from the Nurses’ Health Study and Health Professionals follow-up study); nonetheless, it was based on patients’ self-assessment and self-reporting, and analyzed the impact of the above-mentioned dietary agents only on hypertension incidence, without taking exact BP-value measurements into consideration [7]. The flavonol intake in that study was also calculated on the basis of a general semiquantitative food-frequency questionnaire, and not by the dedicated tool [7]. That is why it could possibly omit some of the subtle relationships between isorhamnetin intake and BP values shown in this study. The products particularly rich in isorhamnetin are onions (white and red), kale, asparagus, elderberry, dill and parsley [6,8].

The presented results revealed also significant differences between male and female participants in terms of the size of the effect of isorhamnetin consumption. These effects were observed only in the male subgroup, while the correlation was not significant in the female subgroup. This is an interesting observation, as in the study by Knekt et al. the effect of flavonoid intake on general CAD mortality was observed in women, but not in men [28]. It is worth noting that hypertension is a CAD risk factor, but in the study by Knekt et al. the direct BP values were not taken into consideration. However, in the studies which analyzed the impact of other flavonol intake (quercetin) directly on hypertension, it was shown that quercetin supplementation results in antihypertensive effects in men [29]. Nonetheless, the present study is the first study which has investigated the antihypertensive potential of dietary isorhamnetin in humans.

The results showed that the main contributors to flavonol intake were onions, tomatoes among the vegetables; blueberries and apples among the fruit; and tea, coffee and wine among the beverages, which matches the observations made in the Zutphen Elderly study [30]. The analysis showed that among these selected products, only onions and tomato mean daily-intakes were significantly correlated with SBP and DBP, respectively. The subgroup analysis confirmed this observation, as the patients with elevated SBP were characterized by significantly lower white onion consumption. Onion extract was already proven to have potential in BP-lowering, although the doses used in the study by Brull et al. and Kalus et al. were higher than reachable in an everyday diet [31,32]. The impact of onion and apple intake on cardiovascular mortality was shown in the study by Knekt et al. and Hertog et al. [28,30]. The subgroup analysis showed that patients with elevated and normal BP also significantly differed in terms of blueberry consumption. This relationship was not presented for berries in the above-mentioned study; however, it could be explained that in the presented study berries were divided into species, and only blueberry intake was taken into consideration, in contrast with the other studies [28]. Even though the studies did not analyze the direct values of BP, it is worth noting that hypertension is a cardiovascular risk factor. The relation between the intake of food rich in flavonoids such as onions, apples or tea, and cardiovascular risk factors (including BP) was also shown in French women, although the authors of the SU.VI.MAX study did not reveal the exact correlations [33]. The important factor could be also the preparation of the meals, including the products mentioned, as boiling could decrease the antihypertensive potential in onions [34].

Apples consumption is popular, due to the beneficial role for health. The term “an apple a day keeps a doctor away” was examined in the course of this study, and it revealed that patients who consume one apple a day or more do not have significantly lower SBP or DBP, compared to the patients with lower apple-consumption. This observation might be caused by the lower border for the minimal apple-consumption impact, as the study investigating the impact of fresh fruit intake and acute coronary syndrome proved that the level of consumption of 25 g/day reduces the risk of acute coronary syndromes [19]. What is more, coffee consumption, which is suggested to elevate BP, did not present such properties in CAD patients. The patients who drink one coffee a day or more did not have significantly higher SBP or DBP compared to the patients who drink less than one coffee a day. This observation matches the results from other studies regarding CVD risk and coffee consumption [20].

On the basis of the presented results, we can suggest that incorporating products such as onion, tomatoes and blueberries into the everyday diet could be possibly beneficial in terms of BP values. Nonetheless, a longer observation on a larger population, or ideally, a controlled prospective study is needed to support this.

In our observational study, the mean daily intake of quercetin was much lower (29.77 [±22.18] mg/day) than the supplementation doses used in randomized controlled studies in humans (50 mg to 730 mg/day), so this may explain why the results from this observation do not match the results from the meta-analysis, which showed that quercetin supplementation could decrease BP values [12]. It is also worth noting that bioavailability from an artificial supplement can differ from that from a dietary source. 

Apart from the mentioned study, there are no other available observational or interventional studies focused on the impact of other flavonols (kaempferol and myricetin) on BP level.

## 5. Conclusions

This study revealed the relationship between long-term dietary isorhamnetin consumption and SBP values in male patients. The correlation was not proved for other flavonols or for DBP. The analysis of specific foods showed that onion, tomato and blueberry intake could impact BP values. This may suggest that a dietary approach which includes a higher intake of products rich in this compound could possibly result in BP lowering.

## Figures and Tables

**Figure 1 nutrients-14-04586-f001:**
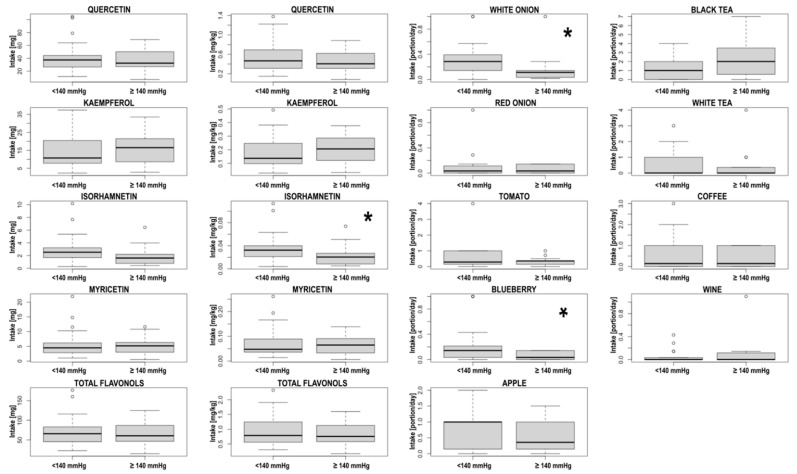
Box-plots presenting differences in flavonol intake and their main dietary source consumption among patients with systolic blood pressure <140 mmHg and ≥140 mmHg (significant differences are marked with *).

**Table 1 nutrients-14-04586-t001:** The correlation between selected flavonol daily intake and systolic and diastolic blood pressure.

	Systolic Blood Pressure			Diastolic Blood Pressure		
	R	95% CI	*p*	R	95% CI	*p*
Quercetin daily intake	−0.23	−0.506; 0.087	0.15	0.05	−0.264; 0.357	0.75
Kaempferol daily intake	0.02	−0.292; 0.331	0.89	0.19	−0.125; 0.477	0.23
Isorhamnetin daily intake	−0.36	−0.602; −0.052	0.02	0.05	−0.263; 0.359	0.74
Myricetin daily intake	−0.08	−0.379; 0.241	0.64	0.09	−0.230; 0.388	0.59
Total flavonol daily intake	−0.18	−0.462; 0.143	0.28	0.10	−0.222; 0.396	0.55
Quercetin daily intake/body mass	−0.28	−0.542; 0.037	0.08	0.04	−0.272; 0.350	0.79
Kaempferol daily intake/body mass	−0.05	−0.354; 0.268	0.77	0.18	−0.138; 0.466	0.26
Isorhamnetin daily intake/body mass	−0.38	−0.617; −0.076	0.02	0.07	−0.250; 0.370	0.68
Myricetin daily intake/body mass	−0.13	−0.426; 0.187	0.41	0.07	−0.247; 0.374	0.67
Total flavonol daily intake/body mass	−0.23	−0.503; 0.091	0.16	0.09	−0.232; 0.387	0.60

**Table 2 nutrients-14-04586-t002:** The correlation between selected flavonols daily intake and systolic and diastolic blood pressure in male participants.

	Systolic Blood Pressure			Diastolic Blood Pressure		
	R	95% CI	*p*	R	95% CI	*p*
Quercetin daily intake	−0.32	−0.661; 0.128	0.16	−0.14	−0.542; 0.307	0.53
Kaempferol daily intake	0.11	−0.340; 0.515	0.64	0.01	−0.422; 0.441	0.96
Isorhamnetin daily intake	−0.65	−0.844; −0.302	0.001	−0.12	−0.515; 0.340	0.64
Myricetin daily intake	−0.06	−0.478; 0.383	0.80	−0.11	−0.518; 0.336	0.63
Total flavonol daily intake	−0.27	−0.626; 0.187	0.24	−0.13	−0.530; 0.323	0.58
Quercetin daily intake/body mass	−0.27	−0.628; 0.183	0.24	−0.08	−0.497; 0.361	0.72
Kaempferol daily intake/body mass	0.16	−0.292; 0.554	0.49	0.11	−0.338; 0.517	0.63
Isorhamnetin daily intake/body mass	−0.63	−0.837; −0.280	0.002	−0.07	−0.491; 0.369	0.75
Myricetin daily intake/body mass	−0.03	−0.457; 0.405	0.89	−0.08	−0.491; 0.369	0.75
Total flavonol daily intake/body mass	−0.21	−0.585; 0.248	0.37	−0.05	−0.470; 0.392	0.84

**Table 3 nutrients-14-04586-t003:** The correlation between selected flavonol daily intake and systolic and diastolic blood pressure in female participants.

	Systolic Blood Pressure			Diastolic Blood Pressure		
	R	95% CI	*p*	R	95% CI	*p*
Quercetin daily intake	−0.17	−0.580; 0.308	0.49	0.30	−0.180; 0.663	0.21
Kaempferol daily intake	−0.09	−0.521; 0.381	0.72	0.34	−0.136; 0.687	0.16
Isorhamnetin daily intake	0.01	−0.447; 0.461	0.97	0.34	−0.130; 0.691	0.15
Myricetin daily intake	−0.14	−0.556; 0.339	0.58	0.37	−0.101; 0.706	0.12
Total flavonol daily intake	−0.14	−0.556; 0.339	0.58	0.32	−0.157; 0676	0.18
Quercetin daily intake / body mass	−0.27	−0.643; 0.213	0.27	0.27	−0.206; 0.647	0.26
Kaempferol daily intake / body mass	−0.19	−0.591; 0.292	0.44	0.30	−0.179; 0.664	0.21
Isorhamnetin daily intake / body mass	−0.06	−0.502; −0.404	0.80	0.37	−0.105; 0.703	0.12
Myricetin daily intake / body mass	−0.23	−0.618; 0.252	0.35	0.32	−0.161; 0.674	0.19
Total flavonol daily intake / body mass	−0.24	−0.625; 0.242	0.33	0.29	−0.192; 0.656	0.23

**Table 4 nutrients-14-04586-t004:** The correlation between selected product daily intake and systolic and diastolic blood pressure.

	Systolic Blood Pressure			Diastolic Blood Pressure		
	*p*	95% CI	R	*p*	95% CI	R
White onion	0.01	−0.624; −0.088	−0.39	0.87	−0.335; 0.288	−0.03
Red onion	0.15	0.508; 0.084	−0.23	0.34	−0.166; 0.444	0.15
Onion (total)	0.02	−0.616; −0.073	−0.38	0.76	−0.265; 0.357	0.05
Tomatoes	0.31	−0.454; 0.153	−0.17	0.04	−0.581; −0.020	−0.33
Blueberry	0.21	−0.483; 0.116	−0.20	0.79	−0.349; 0.273	−0.04
Apples	0.39	−0.431; 0.181	−0.14	0.68	−0.371; 0.249	−0.07
Black tea	0.58	−0.228; 0.391	0.09	0.22	−0.121; 0.480	0.20
Green tea	0.57	−0.393; 0.225	−0.09	0.25	−0.132; 0.472	0.19
Coffee	0.97	−0.306; 0.317	0.01	0.96	−0.318; 0.305	−0.01
Wine	0.89	−0.337; 0.294	−0.02	0.52	−0.215; 0.409	0.11

**Table 5 nutrients-14-04586-t005:** Differences in flavonol and selected product daily intake in patients with normal systolic blood pressure (<140 mmHg) and elevated systolic blood pressure (≥140 mmHg).

	Systolic Blood Pressure				
	<140 mmHg		≥140 mmHg		
	Mean	SD	Mean	SD	*p*
Quercetin [mg/day]	42.02	±24.81	36.73	±17.24	0.61
Kaempferol [mg/day]	13.91	±8.66	16.14	±8.24	0.39
Isorhamnetin [mg/day]	2.88	±2.20	1.90	±1.53	0.08
Myricetin [mg/day]	5.66	±4.80	5.39	±3.02	0.59
Total flavonols [mg/day]	72.09	±39.77	67.57	±30.94	0.94
Quercetin [mg/kg*day]	0.55	±0.33	0.45	±0.22	0.55
Kaempferol [mg/kg*day]	0.18	±0.12	0.20	±0.10	0.52
Isorhamnetin [mg/kg*day]	0.04	±0.03	0.02	±0.02	0.048
Myricetin [mg/kg*day]	0.07	±0.06	0.07	±0.04	0.94
Total flavonols [mg/kg*day]	0.94	± 0.54	0.83	±0.40	0.68
White onion [portion/day]	0.31	±0.25	0.17	±0.25	0.01
Red onion [portion/day]	0.09	±0.21	0.07	±0.06	0.51
Tomatoes [portion/day]	0.58	±0.82	0.34	±0.27	0.64
Blueberries [portion/day]	0.24	±0.31	0.06	±0.06	0.04
Apples [portion/day]	0.68	±0.51	0.56	±0.47	0.65
Black tea [portion/day]	1.26	±1.22	2.27	±2.03	0.10
Green tea [portion/day]	0.49	±0.77	0.45	±1.08	0.44
Coffee [portion/day]	0.76	±0.88	0.39	±0.47	0.37
Wine [portion/day]	0.05	±0.10	0.11	±0.29	0.82

## Data Availability

The data that support the findings of this study are available from the corresponding author upon reasonable request.

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
