# Peer review of "Dietary Isorhamnetin Intake Is Associated with Lower Blood Pressure in Coronary Artery Disease Patients"

_nutrients, 2022, doi:10.3390/nu14214586_

Round 1

Reviewer 1 Report

1. What percentage of the total population in Japan has hypertension? You can't judge whether there are too many or too few just by looking at the number of people.

2. Regarding the antioxidant effect of isorhamnetin, I would like to know which type of active oxygen is removed to regulate blood pressure.

3. Mann-Whitney's U-test was used for statistical processing in this study, but why did you not use the F-test and the student-T-test? (because it is an intervention study)

Author Response

Dear Reviewer,

Thank you very much for your time, effort and valuable comments to our manuscript. We corrected the manuscript according to your suggestions and we believe that it deeply improved its quality. Below you can find our responses to your comments. Thank you for this opportunity.

1. What percentage of the total population in Japan has hypertension? You can't judge whether there are too many or too few just by looking at the number of people.

- We kindly assume that the Reviewer meant population of Poland, which was the origin of the study. The information about the prevalence in Poland and worldwide was added in lines 33-34.

2. Regarding the antioxidant effect of isorhamnetin, I would like to know which type of active oxygen is removed to regulate blood pressure.

- The biological effects of isorhamnetin are still under investigations, however the results by other authors suggest that these mechanisms are beyond ROS scavenging. This information about potential mechanism of isorhamnetin biological effects was added in lines 183-186.

3. Mann-Whitney's U-test was used for statistical processing in this study, but why did you not use the F-test and the student-T-test? (because it is an intervention study)

- The groups analyzed in that part of this observational study were independent groups and Shapiro-Wilk test revealed that the data distribution in these particular subgroups was not normal. That is why, Mann-Whitney's U-test was used for this analysis.

Reviewer 2 Report

The manuscript "Dietary isorhamnetin intake is associated with lower blood 2 pressure in coronary artery disease patients" by Popiolek-Kalisz et al. presents interesting data on relationship between isorhamnetin and blood pressure levels in CVD patients.

Below are comments that will benefit the manuscript and the discussion 

1. English language revision is needed all throughout the manuscript.

Some words need correction on multiple places "None the less" is one word "Nonetheless"

Abstract Line 25: replace the word "reach" with "rich" (isorhamnetin-rich)

2. As authors acknowledge, isorhamnetin is less well known than other flavonols quercetin and kaempferol. A short information on this flavonol could be included in the introduction. 

3. Line 74: please define "long term". For how many days the participants were asked to enter the food frequency questionnaire? 

4. Were there any differences between male vs. female participants in terms of the size of the effect?

Author Response

Dear Reviewer,

Thank you very much for your time, effort and valuable comments to our manuscript. We corrected the manuscript according to your suggestions and we believe that it deeply improved its quality. Below you can find our responses to your comments. Thank you for this opportunity.

The manuscript "Dietary isorhamnetin intake is associated with lower blood pressure in coronary artery disease patients" by Popiolek-Kalisz et al. presents interesting data on relationship between isorhamnetin and blood pressure levels in CVD patients. Below are comments that will benefit the manuscript and the discussion 

  1. English language revision is needed all throughout the manuscript.Some words need correction on multiple places "None the less" is one word "Nonetheless". Abstract Line 25: replace the word "reach" with "rich" (isorhamnetin-rich)

- The manuscript underwent extensive English revisions and the suggested corrections were made.

  1. As authors acknowledge,isorhamnetin is less well known than other flavonols quercetin and kaempferol. A short information on this flavonol could be included in the introduction. 

- This information was added in lines 43-64.

  1. Line 74: please define "long term". For how many days the participants were asked to enter the food frequency questionnaire? 

- The information was gathered for one year period. This information was added in the line 87.

  1. Were there any differences between male vs. female participants in terms of the size of the effect?

- That was a very valuable suggestion. We conducted subgroup analysis which revealed that the effect was present only in male participants. This information was added to abstract (lines 18-19), results (lines 139-148) and discussion (lines 204-214). Thank you once again for this suggestion.